## RESEARCH ARTICLE

# Taxonomic diversity and functional adaptations indicated by the rhizospheric soil microbiome derived from Turkish wheat fields

Gülce Güralp[1], Sena Nur Acet[2], Jana al-Khodor[1], Özlem Akkaya[2,3], M. G. Şeker[2,3,4], Veysel Süzerer[5], Y. Özden Çiftçi[2,3,4] and Stuart J. Lucas[1,6,*]

## ABSTRACT

Optimization of the soil microbiome is a promising strategy to support sustainable crop production. With the goal of developing novel bio-fertilizers for wheat cultivation, we collected fresh soil samples from ten different fields representing wheat production regions in Türkiye. Wheat seedlings (*Triticum turgidum* ssp. *durum*) were cultivated in each soil and at the three-leaf stage, DNA was isolated from the rhizospheric soil associated with each plant and the bacterial microbiome composition determined by 16S metabarcoding. Long-read sequencing was used to maximize resolution, and 1269 high-quality operational taxonomic units (OTUs) were identified. Comparisons of wheat and non-wheat rhizospheric soil identified 77 OTUs that were enriched in the wheat rhizosphere, several belonging to taxa that have previously been described as plant growth-promoting rhizobacteria. Furthermore, 209 OTUs were present in all ten wheat fields sampled, indicating that they may be ubiquitous in wheat-growing regions of Türkiye; a subset of these were also reported in wheat rhizospheric soil from other countries. Additional taxa were shown to be enriched based on local soil conditions such as pH and macronutrient content. These findings shed light onto the essential composition of the wheat rhizospheric microbiome, which provides a foundation for the development of locally adapted bio-fertilizers.

KEY WORDS: Soil microbiome, Rhizosphere, Wheat, Plant growth-promoting rhizobacteria, Türkiye

## INTRODUCTION

Wheat (*Triticum* spp.) continues to be the world's most extensively grown, widely consumed and traded crop, with worldwide utilization of approximately 800 million tonnes annually (FAO, 2025). Therefore, wheat production makes a critical contribution to global and regional food security. Over 60% of wheat is cultivated in rainfed fields, increasing sensitivity to the effects of climate change (Dadrasi et al., 2023). At the same time, there is a need to increase production to keep pace with the growing world population, while also reducing agricultural inputs such as fertilizer and pesticide. Türkiye is one of the world's top ten wheat producers and a country for which these concerns are highly relevant, as wheat is a staple part of the diet and rising temperatures are projected to have a significant negative impact in all production regions (Demirhan and Bayraktar, 2025).

In this context, the last decade has seen an increasing interest in the role of plant-associated microbiota in influencing crop yield, health and resilience. While individual microbes may be pathogenic, as a whole plant microbiome communities are generally believed to play a beneficial role. In particular, the root rhizosphere is the largest plant microbial habitat and consists of a complex domain where plant exudates, microorganisms and soil interact to regulate nutrient availability, recycle waste products and support adaptation to environmental stress conditions (Ling et al., 2022). Therefore, deciphering the rhizosphere microbiome has great relevance both to understanding plant stress responses and as a means for monitoring soil fertility and biodiversity (Guerrieri et al., 2021). Furthermore, deliberate modification of the rhizosphere microbiome composition is proposed as a promising strategy for promoting sustainable crop production, such as by reducing the dependence on chemical fertilizers or helping protect against pests and pathogens (Afridi et al., 2022). For example, studies of the tomato rhizosphere microbiome identified a flavobacterium that confers resistance to infection by the soil-borne phytopathogen *Ralstonia solanacearum* (Kwak et al., 2018). Similarly, studies in various crops have identified specific rhizospheric bacterial genera associated with abiotic stress tolerance (Fan et al., 2023; Zheng et al., 2021).

Microbiome analysis has been greatly accelerated by the development of high-throughput DNA sequencing technologies, which provide an unprecedented window into the diversity of natural microbial communities through metagenomics methods (Priya et al., 2021). One widely used strategy for surveying microbiome diversity is 'meta-barcoding', in which a ubiquitous gene – usually the 16S small subunit of the ribosomal RNA in the case of bacteria – is amplified from an environmental sample by PCR and then sequenced (Bulgarelli et al., 2012). The resulting wealth of amplicon sequences are analysed using appropriate statistical methods to obtain a model of the community from which they were derived (Shelton et al., 2016). The ability to harvest DNA sequences from microbes that have extremely low abundance, or are refractory to standard culture methods, has revealed that previously unknown strains comprise the majority in many environmental samples (Hug et al., 2016). This provides great potential for novel discoveries alongside significant challenges in data interpretation, as these new strains do not have a species definition (Sanford et al.,

[1]Sabanci University, Faculty of Engineering and Natural Sciences, 34956 Tuzla, Istanbul, Türkiye. [2]Gebze Technical University, Department of Molecular Biology & Genetics, 41400 Gebze, Kocaeli, Türkiye. [3]Gebze Technical University, Central Research Laboratory Research and Application Center, GTU MAR, 41400 Gebze, Kocaeli, Türkiye. [4]Gebze Technical University, Smart Agriculture Research and Application Center, GTU ATAM, 41400 Gebze, Kocaeli, Türkiye. [5]Bingöl University, Vocational School of Health Services, Department of Pharmacy Services, 12000 Bingöl, Türkiye. [6]Sabanci University Nanotechnology Research and Application Center (SUNUM), 34956 Tuzla, Istanbul, Türkiye.

*Author for correspondence (slucas@sabanciuniv.edu)

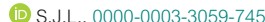 S.J.L., 0000-0003-3059-7453

2021). Therefore, historically most datasets were reported at the level of operational taxonomic units (OTUs), grouping together DNA barcodes based on similarity [typically ≥97% sequence identity (Bulgarelli et al., 2012)]. Each OTU can be considered a 'species hypothesis' but may actually combine sequences from multiple species, while even individuals from the same species may differ in their functional gene content (Lladó Fernández et al., 2019). Given these uncertainties, methods for characterising plant microbiomes are evolving almost as rapidly as the communities themselves. Long-read meta-barcoding using Nanopore or PacBio platforms can incorporate the whole 16S gene into a single amplicon, improving resolution; however, they currently have higher per-base error rates than Illumina, while their data analysis pipelines are still under development (Santos et al., 2020).

Even so, meta-barcoding studies have already provided valuable insights into plant-microbiome interactions for many crops, including wheat. The composition of the wheat rhizosphere microbiome has been shown to alter in response to tillage practices, crop rotation and the use of conventional or organic agricultural practices (Simonin et al., 2020; Yin et al., 2010). Moreover, the microbiome also evolves during the course of the season, with the community established during vegetative growth becoming more diverse as the plant reaches reproductive maturity (Donn et al., 2015). Mahoney et al. (2017) compared nine different winter wheat cultivars planted on a farm in Washington, USA, and identified 962 OTUs that were found in 95% of rhizosphere samples, proposing that they made up a core microbiome across the cultivars. Developing this concept, soils from four countries in Europe and Africa were collected and used to cultivate the same three winter wheat genotypes under controlled conditions (Simonin et al., 2020). This demonstrated that the majority of micro-organisms present in the wheat rhizosphere depend on the soil of origin, but 57 taxa were present in all the countries sampled and might be considered a 'true core microbiome'.

These reports demonstrated that the wheat rhizosphere microbiome varies much more with geography than the wheat cultivar planted. However, little is known about the microbiome composition in many of the world's 'bread baskets', including Türkiye. In this study, we sampled post-harvest soils from ten different farms, spanning the geoclimatic diversity of wheat production areas in Türkiye. Each soil was used to cultivate new wheat plants in a growth chamber, from which rhizospheric soil was collected to characterize the bacterial microbiome by 16S metabarcoding. We adopted a long-read amplicon sequencing approach using the Oxford Nanopore Technologies MinION platform; this allows the complete 16S gene to be sequenced from a single molecule, allowing more comprehensive assessments of diversity than shorter reads that only cover some of the 16S variable regions (Veselovsky et al., 2025). Therefore, we aimed to evaluate the utility of this technology while addressing the following questions: to what extent does the wheat rhizosphere microbiome in Turkish soils depend on soil chemistry and location? Is it possible to define a core rhizosphere microbiome, and is its composition similar to that reported from other countries? Answering these questions will give us greater potential to identify bacterial taxa that contribute to wheat production and could be deployed as part of deliberate soil improvement strategies.

## RESULTS

### Soils used for wheat cultivation show variable nutrient profiles

Soil was collected from 10 different locations as shown in Fig. 1, and results of physical and chemical analysis are summarized in Table 2. All of the soils tested had low salinity, and the majority had pH close to neutral, while samples Tek2, Diy1 and Diy2 were mildly acidic. As expected, lower pH correlated with lower lime content and higher soil water capacity, although all values were within ranges considered appropriate for wheat cultivation. However, several samples (Ank1, Ank2, Tek2 and Siv2) had relatively low total nitrogen; the first three of these were also deficient in available phosphate, which was reflected in reduced plant height at the three-leaf stage. Both soil acidity and macronutrient levels could affect the rhizospheric microbiome, which could assist plant growth, for example by mobilizing scarce nutrients.

### Long-read metabarcoding characterises the bacterial rhizosphere microbiome

Wheat plants grown in each of the ten soil samples were harvested at the three-leaf stage to collect rhizospheric soil. In addition, two non-wheat rhizosphere soil samples were included for comparison. From each sample, total DNA was isolated and 16S rRNA gene amplified and sequenced as described in the Materials and Methods. To test for bias introduced during library preparation, the sequencing protocol was carried out three times with the barcode adapters assigned to different samples in each run. All runs gave similar results in terms of the abundance of OTUs detected and clustering of samples (see Figs S1 and S2), even though there was some variation between runs and between samples in sequencing depth. The peak read length was ~1.5 kb in all cases, demonstrating that the 16S (V1-V9) amplicon was sequenced mostly as a single read. Run 3 was the most productive run (1.23 million passed reads, more than twice as many as the other runs) and also had the most even distribution of reads between the 12 samples and so was used for the analyses below.

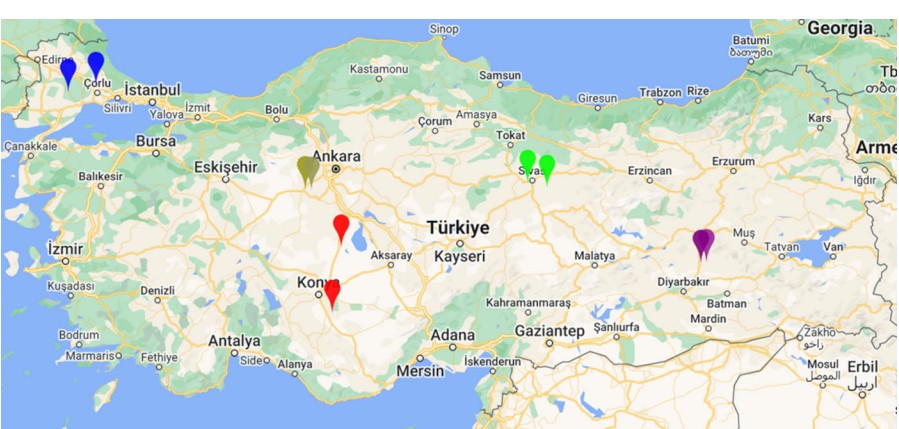

**Fig. 1. Map of ten soil sampling locations in five provinces of Türkiye.** Tekirdağ (blue, northwest corner) and Diyarbakır (purple, southeast) are low altitude plains with wet winters and warm, dry summers. The other sites are all on the Anatolian plateau with a cool, drier continental climate.

**Table 1. Soil sampling locations and field characteristics**

| Province | Sample | Location | Co-ordinates | Crop rotation | Soil type* | ID code |
|---|---|---|---|---|---|---|
| Ankara | Field 1 | Polatlı/Kargalı | 39°36′ N 32°13′ E | None | Rough, broken land; brown soil | **Ank1** |
| | Field 2 | Polatlı/Beyobası | 39°35′ N 32°20′ E | None | | **Ank2** |
| Diyarbakır | Field 1 | Yolçatı/Lice | 38°24′ N 40°41′ E | None | Rough mountainous; brown forest soil | **Diy1** |
| | Field 2 | Lice | 38°25′ N 40°42′ E | None | | **Diy2** |
| Konya | Field 1 | Çumra | 37°34′ N 32°47′ E | None | Alluvial soil | **Kon1** |
| | Field 2 | Altınekin | 38°29′ N 32°55′ E | None | Reddish brown soil | **Kon2** |
| Sivas | Field 1 | Merkez/Serpincik | 39°42′ N 36°55′ E | None | Rough, broken land; brown soil | **Siv1** |
| | Field 2 | Merkez/Esenyurt | 39°37′ N 37°20′ E | None | | **Siv2** |
| Tekirdağ | Field 1 | Muratlı/İnanlı | 41°17′ N 27°48′ E | Maize | Non-calcic brown soil | **Tek1** |
| | Field 2 | Hayrabolu/Delibedir | 41°10′ N 27°12′ E | Maize | Grumusol soil | **Tek2** |

*Soil classifications from EUDASM (EU Digital Archive of Soil Maps, https://esdac.jrc.ec.europa.eu).

In order to assess how fully our data represents the wheat rhizosphere microbiome, rarefaction curves were plotted and within-sample diversity (Alpha diversity) calculated for all samples (Fig. 2). These rarefaction curves approached saturation with increasing read depth (Fig. 2A), suggesting that our data denotes the large majority of taxa present in each sample. The number of observed OTUs in samples from Diyarbakır (Fig. 2A, orange traces) was lower than from other locations, but these differences were not statistically significant. The Shannon diversity index was also similar for all locations (Fig. 2B), indicating that the differences in observed OTUs correspond to taxa with low abundance.

In all samples a proportion of reads were given higher level taxonomic classifications (phylum, class or family) but could not be assigned to a specific genus or species, an expected result owing to the error rate of the sequencing chemistry. Therefore, for quantification of differentially abundant taxa, OTUs that were not classified at least at the family level (35.8% of total read counts) were eliminated from the data. Next, to avoid false positives resulting from random sequencing errors, those represented by fewer than three reads/sample in at least two different samples were discarded, leaving a total of 1269 OTUs across all samples (full OTU counts and taxonomic assignments are in Table S1).

Using the filtered data, dissimilarity between samples was visualised by multi-dimensional scaling based on Jaccard and Jensen-Shannon divergence metrics (Fig. 3A). Samples originating from the same province are usually clustered together, while the two non-wheat soils were clearly separated from the wheat rhizospheric samples (P<0.05 by ANOSIM). Depending on the metric used, the acid soils also clustered separately from the neutral soils, although this difference was not statistically significant.

Compositional differences between the wheat rhizospheric and non-wheat soil samples were quantified and 77 differentially abundant OTUs were identified (FDR adjusted P-value <0.05; Fig. 3B), of which 72 showed increased abundance in wheat rhizospheric soil. Strikingly, 30 of these were assigned to *Pseudomonas*, although the differentially abundant species from this genus varied from sample to sample. Therefore, when the read counts were agglomerated by genus, *Pseudomonas* was not indicated as differentially abundant (as some *Pseudomonas* species were also abundant in non-wheat soils). On the other hand, after agglomeration 15 genera and six families for which genus could not be determined were consistently more abundant in the wheat rhizosphere than non-wheat soil (Table S2), with the most significant (q-value <0.01) being *Methylotenera*, *Pantoea* and *Rhodoferax*, each of which are represented by multiple species (Fig. 3B). This suggests that, along with a sub-group of *Pseudomonas*, these genera have an important functional role in the wheat rhizosphere.

The effect of soil conditions on the microbiome were illustrated by comparing the acid and neutral soils (Fig. 3C); four OTUs assigned to *Massilia* and *Rhodanobacter* were consistently more abundant in low pH rhizospheric soil, which may suggest that they help to mobilize nutrients under acid conditions. In contrast, *Arenimonas*, *Pseudoxanthomonas* and *Steroidobacter* were more abundant in neutral soil.

## Evidence for a core wheat rhizosphere microbiome

In order to define a 'Turkish wheat rhizosphere core microbiome', we took a further subset of the data consisting of OTUs that were present in wheat soil samples from all ten locations (Table S3). This core microbiome included 209 OTUs representing 38 different taxonomic families, although it was notable that the Comanomonadaceae, Oxalobacteraceae and Xanthomonadaceae families were by far the most abundant in terms of both read count and OTU diversity (Fig. 4A). The relative abundance of these

**Table 2. Soil sample properties**

| Sample ID | pH | EC (mS/cm) | Total soluble salt (%) | Soil water capacity (%) | Organic content (%) | Lime content (%) | Total N (%) | Available phosphate (kg/da) | Total K (kg/da) | Average plant height (cm)* |
|---|---|---|---|---|---|---|---|---|---|---|
| Kon1 | 7.06 | 1.35 | 0.046 | 52.8 | 2.18 | 23.48 | 0.235 | 20.95 | 241.8 | 27.72±5.30^c |
| Kon2 | 7.22 | 0.74 | 0.024 | 51.7 | 2.24 | 17.80 | 0.249 | 35.95 | 260.4 | 44.31±1.23^a |
| Tek1 | 6.83 | 1.41 | 0.056 | 61.6 | 1.39 | 2.70 | 0.165 | 11.28 | 43.2 | 39.87±0.28^a |
| Tek2 | 5.81 | 0.43 | 0.017 | 60.5 | 1.42 | 2.14 | _0.124_ | _4.24_ | 36.6 | 35.03±1.63^b |
| Ank1 | 7.20 | 0.96 | 0.031 | 50.6 | _0.20_ | 10.68 | _0.095_ | _0.06_ | 30.0 | 36.76±1.26^a |
| Ank2 | 7.33 | 0.82 | 0.028 | 52.8 | 1.79 | 14.24 | _0.103_ | _0.06_ | 37.2 | 38.50±0.60^b |
| Siv1 | 7.08 | 0.69 | 0.022 | 49.5 | 2.10 | 22.78 | 0.204 | 7.44 | 57.0 | 43.49±1.56^a |
| Siv2 | 7.15 | 0.65 | 0.02 | 49.5 | 1.79 | 21.36 | _0.116_ | 20.04 | 31.5 | 40.60±3.72^a |
| Diy1 | 5.67 | 0.94 | 0.037 | 61.6 | 2.41 | 2.56 | 0.213 | 23.30 | 51.0 | 37.91±3.25^ab |
| Diy2 | 5.84 | 1.07 | 0.051 | 74.8 | 1.90 | 2.84 | 0.199 | 9.50 | 89.4 | 41.59±2.85^a |

Values indicating a nutrient deficiency are underlined.
*Groups (n=3) were compared by one-way ANOVA with Tukey's post-tests. Different letters indicate statistically significant differences between groups (P<0.05).

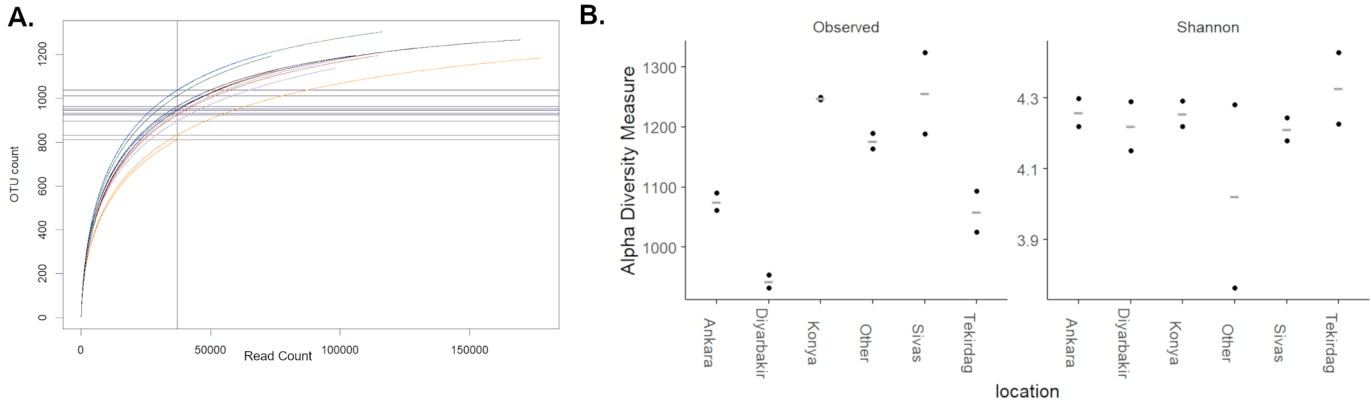

**Fig. 2. Within-sample diversity of 16S microbiome data.** (A) Rarefaction curves plotted by random sub-sampling of each dataset without replication. Vertical line shows the crossing point for each curve rarefied to the level of the smallest sample (Diy1; 37,052 reads). (B) Observed (left) and Shannon (right) diversity measures calculated after rarefaction to 37,000 reads. Samples are plotted individually (*n*=2) along with mean values for each location (−).

dominant families was approximately similar in different samples. However, single taxa usually had a much more varied distribution. For example, the single most abundant OTU across the dataset, assigned to *Lysobacter terricola* (Xanthomonadaceae), had a

relative abundance by read count varying from 0.1% (Siv1) to 10.7% (Diy2) depending on the sample.

Recognising this diversity, network modelling was used to identify clusters of OTUs and hub taxa within the microbiome

**Fig. 3. Microbiome differences between samples.** (A) Beta diversity calculated using Jaccard (top) and Jensen-Shannon (bottom) distances, illustrated by MDS. For soil pH, nd=not determined (the case for the non-wheat soils). (B) Heatmap of differentially abundant OTUs between wheat rhizospheric (*n*=10) and non-wheat (*n*=2) soils, using ancombc2 method with BY correction for multiple comparisons (q<0.05). (C) Heatmap of differentially abundant OTUs between acidic (*n*=3) and neutral (*n*=7) soils, by the same method, non-wheat soils were not included in the statistical comparison but the relative abundances of the identified OTUs in these soils are also shown in the heatmap. Mean-scaled Abundance: counts were normalized by the total counts from each sample, and the fold-change values within each row scaled to the mean before plotting.

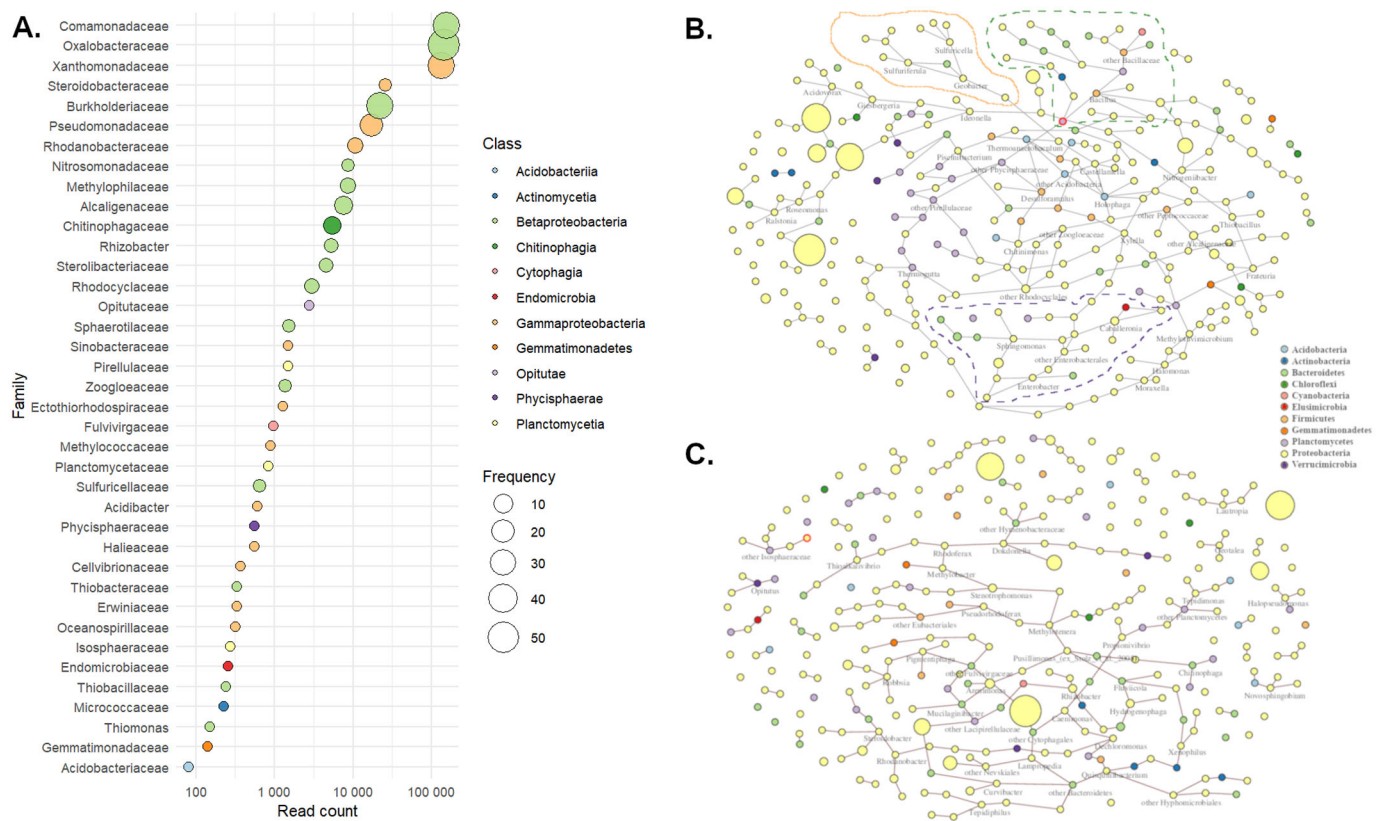

**Fig. 4. Summary and modelling of the Turkish wheat rhizospheric core microbiome derived from all samples (*n*=10).** (A) Bubble plot of the 38 taxonomic families in the core microbiome in descending order of relative abundance. Circles are coloured by taxonomic class and sizes indicate the number of different OTUs identified from each family. (B,C) Network plots of inferred positive (B) and negative (C) interactions within the rhizosphere bacterial microbiome, generated using the SpiecEasi algorithm (see Materials and Methods for statistical parameters). Node colours indicate taxonomic phylum, and dotted lines highlight clusters of interacting taxa. For readability, genus names are given only for nodes with four or more nearest neighbours.

(see Materials and Methods). The most abundant taxa were generally not central to the network model (Fig. 4B,C), suggesting that despite their prevalence they do not determine bacterial community structure. The six hub taxa identified all belonged to different genera (*Holophaga*, Xylella, *Bacillus*, *Thiobacillus*, *Ideomonas* and *Enterobacter*) and had low to moderate relative abundance in all samples. Several clusters were identified in the positive interaction network, for example around hubs such as *Bacillus* and *Enterobacter* (Fig. 4B, green dotted and purple dotted lines, respectively). On the other hand, the sub-cluster around *Geobacter* (Fig. 4B, orange line) contains no hub taxa, but several species reported to be capable of iron reduction. *Geobacter* and its syntrophic bacterial partners are known to have an important impact on soil chemistry, recycling organic molecules while reducing insoluble Fe(III) to soluble Fe(II) (Lovley et al., 2011). Therefore, this sub-cluster may represent 'accessory' taxa that help supply wheat roots with iron in a bioavailable form. Notably, some of the taxa that were significantly more abundant in wheat compared to non-wheat soils, such as *Methylotenera* and *Rhodoferax*, had primarily negative interactions in the rhizosphere microbiome network model (Fig. 4C). It has been reported that negative interactions within microbiome networks play an important role in network robustness, e.g. by keeping detrimental species in check (Kajihara et al., 2025 preprint). However, inferences from correlation must be regarded as tentative, and should be confirmed by functional studies using these specific taxa.

## Geographical variation in the wheat rhizosphere microbiome is relatively limited

In order to determine how much local geography determines the wheat rhizospheric microbiome, the non-wheat samples were excluded from the dataset, and each province treated as a separate group for multi-directional comparisons using ancombc2 (see Materials and Methods). This algorithm first checks for the presence and absence of taxa in each group; of the 1269 OTUs included in the initial analysis, 922 (72.7%) were present in all five provinces, with most of the remainder were found in 4/5 provinces (Fig. 5A). This demonstrates a high level of consistency in the qualitative composition of the wheat rhizospheric microbiome across soil conditions, although the relative abundance of taxa may vary. From the 922 common OTUs, 86 were found to be differentially abundant taxa (DATs) between two or more provinces (Fig. 5B; Table S4), belonging to 38 genera (Fig. 5C). The largest number of differences were found between Sivas and other provinces (62, 45, 45 and 73 DATs compared to Ankara, Diyarbakır, Konya and Tekirdağ, respectively), while there were only 12 DATs between Ankara and Konya. This may reflect the fact that the samples from Sivas had the highest observed microbial diversity, particularly of taxa present at low abundance (Fig. 2B), while the sites in Ankara and Konya were closest to each other geographically and have the most similar climate (Iyigun et al., 2013). More broadly, although samples from the same province resemble each other more than those from other provinces (Fig. 5C), most DATs were either distributed across several provinces or strongly associated with a single sampling site;

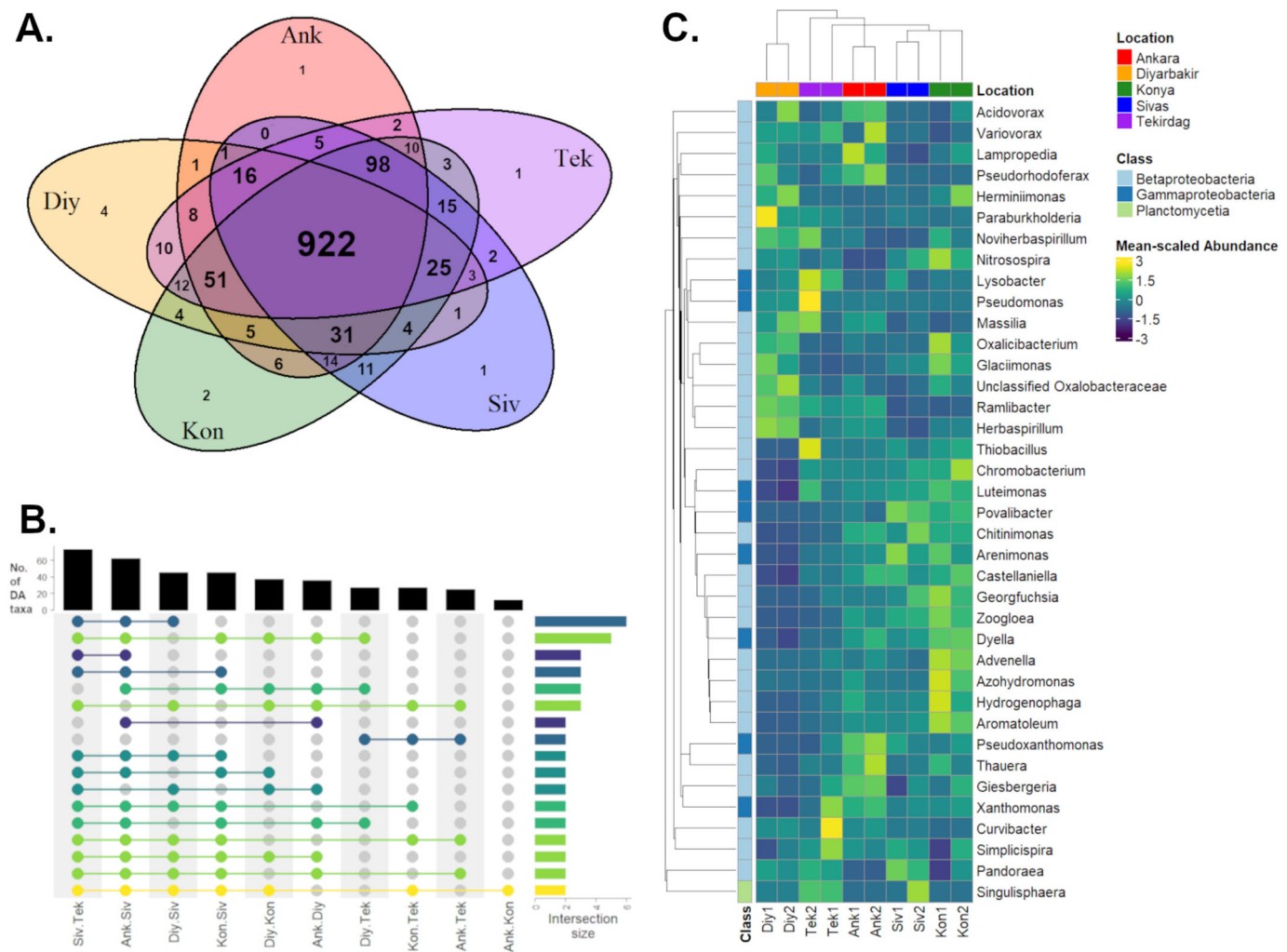

**Fig. 5. Distribution of rhizosphere bacterial taxa across five different provinces.** (A) Venn diagram showing the number of taxa identified in common between samples from different provinces of Türkiye ($n$=2 per province). (B) Distribution of 86 DATs determined by 'ANCOMBC2 global' test, followed by multiple pairwise tests to specify which pairs of provinces differed significantly from each other, with the Holm correction for family-wise error rates (see Materials and Methods). The black bar plot (top) indicates the number of DATs determined between each pair of provinces. Coloured bars (right) indicate the number of DATs with significant differences in comparisons between several pairs of provinces; lines and circles show the intersection between multiple comparisons, coloured by frequency (dark blue -> yellow = fewest -> most pairwise comparisons). For simplicity, only intersections represented by two or more taxa are shown. (C) Heatmap showing relative abundance of all the DATs in each province ($n$=2), agglomerated to the genus level. Mean-scaled Abundance: counts were normalized by the total counts from each sample, and the fold-change values within each row scaled to the mean before plotting.

for example, *Thiobacillus* was highly abundant in Tek2 but not Tek1, while the opposite is true for *Curvibacter*. Taken together, this suggests that most taxa in the Turkish wheat rhizosphere microbiome are not strongly correlated with regional geography, but a minority of taxa are strongly affected by local environmental conditions at a smaller scale.

With this in mind, the possible influence of soil nutrients on the rhizosphere microbiome was evaluated by dividing the soils into three groups; 'poor' soils (Ank1, Ank2, Tek2) were those with low total N and phosphate availability, resulting in reduced growth of test wheat plants (Table 2). 'Moderate' soils (Diy2, Siv2, Tek1) had adequate macronutrient levels, but not as uniformly high as the 'good' soils (Diy1, Kon1, Kon2, Siv1). Only three genera showed a statistically significant and consistent trend across the three groups; the abundance of *Aquicella* and *Legionella* decreased with reducing nutrient levels, suggesting that they may be sensitive to the availability of nitrogen and/or phosphate (Fig. 6). On the other hand,

the abundance of *Stenotrophomonas* increased as nutrient availability decreased.

## DISCUSSION
### Limitations and outputs of our meta-barcode sequencing strategy
Meta-barcoding is an increasingly popular method for evaluating microbiome composition, but there is not yet a consensus on which protocols offer the best balance of sensitivity and accuracy. Alongside the advantage of generating a single amplicon from the 16S gene, the major limitation of the MinION v9 chemistry used here is its high per-base random error rate (~10%), necessitating specialized data processing (Santos et al., 2020) and introducing a risk that species diversity can be overestimated. For example, a previous study using the same chemistry for a mock bacterial community of 15 species detected over 50 OTUs; the excess 'species' belonged to the genera present in the community and were

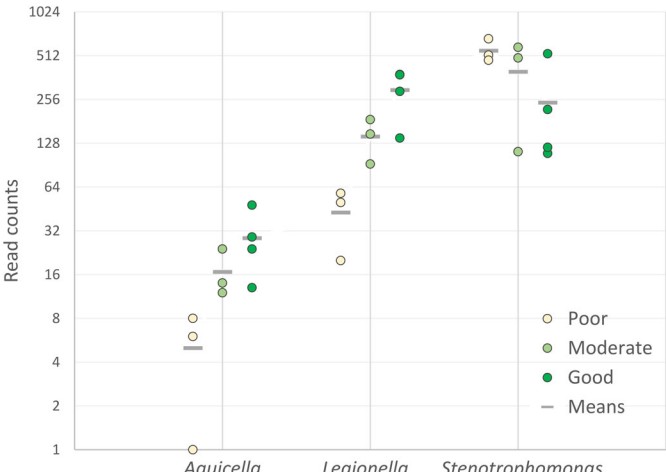

**Fig. 6. Per sample abundance of bacterial genera whose abundance showed a statistically significant correlation with soil nutrients.** 'ANCOMBC2 trend' function with alpha=0.05, Holm-Bonferroni correction, and 100 bootstraps. Read counts for each sample are shown, along with the means for each soil nutrition type: 'Poor' ($n$=3), 'Moderate' ($n$=3), 'Good' ($n$=4).

only detected at low read depths, suggesting that they reflect sequencing errors and/or intra-species diversity within the 16S rRNA gene pool (Lemoinne et al., 2024). Sequencing mock communities can identify technical biases arising during PCR amplification and library preparation; however, other studies have highlighted that they do not account for variability in DNA isolation and amplification efficiency from real soil samples (Manter et al., 2024).

In this study, we minimized inter-sample variability by maintaining consistent isolation and amplification conditions, we also sequenced PCR-negative controls to confirm that there was no bacterial contamination during library preparation. We controlled for over-assignment of OTUs by eliminating those that had <5 reads in total or were only detected in a single sample. However, species-level assignments must still be regarded as tentative, especially for low abundance taxa. We also discarded OTUs that could only be assigned to broader taxonomic levels (order, class, phylum), on the assumption that they represent the same taxa as those which were precisely assigned, but with more sequencing errors. There remains a risk of systematic bias during PCR amplification and sequencing, but this is assumed to be constant for all samples, so inter-sample differences should reflect those in the sampled microbiome communities. With these caveats in mind, we carried out a comprehensive analysis of the bacterial wheat rhizosphere microbiome (Fig. 2A). Based on previous studies, this is expected to contain a subset of the microbial diversity in the surrounding bulk soil, enriched for certain taxa such as Bacteroidetes and Proteobacteria (Ling et al., 2022), which were also the most abundant phyla detected in our dataset. We highlighted differences between the wheat rhizospheric microbiome and two rhizosphere samples collected from other crops grown in different locations, and while some differences may be due to location-specific variation, 72 OTUs were consistently more abundant in the wheat rhizosphere samples than controls (Fig. 3B).

### What is really 'core' for the wheat rhizosphere microbiome?
The concept of a 'core microbiome' was first introduced for human microbiota in 2007, meaning taxa that were common to most or all

samples (Turnbaugh et al., 2007), on the hypothesis that ubiquitous microbes must have an important functional contribution. This idea has been widely adopted in agroecosystems where optimizing microbiome function is a high priority. However, there are no universally accepted criteria for determining which taxa are part of the 'core' (Dong et al., 2021). For example, Mahoney et al. (2017) reported a core microbiome of 962 OTUs that were common to 95% of rhizosphere samples from nine different wheat cultivars, but all grown in the same environment; subsequent studies have shown that the local soil environment impacts microbiome community membership more than wheat genotype (Simonin et al., 2020). We also defined the core microbiota of the wheat rhizosphere in Turkish soils using two criteria: ubiquity, i.e. presence in all samples, along with an abundance cutoff of at least 0.01% of all fully assigned reads (Fig. 4A). The previous study identified 30 bacterial ESVs (exact sequence variants) that were present in wheat rhizosphere samples from eight countries in Europe and Asia (Table 2 in Simonin et al., 2020), of these, 14 were also present in our core microbiome (Table S3), while a further four were found in 9/10 Turkish wheat rhizospheric soils. The clearest difference between the two studies was the low prevalence of alphaproteobacteria in our dataset, in which beta- and gammaproteobacteria make up the large majority of the core microbiome. In contrast, a recent study of wheat rhizosphere soil samples from India described a 'core microbiome' of 20 ubiquitous bacterial genera (Kumar et al., 2023) of which only *Massilia* and *Flavisolibacter* were also identified both in this study and Simonin et al. (2020). Some of these differences may arise from methodological biases in each study, to address this it has been suggested that reference soils should be used to provide inter-laboratory comparisons (Manter et al., 2024). However, the differences also indicate real biological variations between the soils or wheat genotypes used, for example, the rhizosphere microbiomes of elite wheat cultivars are reported to be less diverse than those of ancient varieties (Jacquiod et al., 2022), suggesting that different microbial taxa could have been selected for in different locations. Therefore, a thorough meta-analysis is needed to identify which taxa are truly ubiquitous in the wheat rhizosphere – or whether a different definition of the core microbiome, perhaps based on molecular functions rather than taxonomy, would be more informative.

### The importance of regional versus local environment
All five provinces used for sampling in this study represent different climate zones of Türkiye, except for Ankara and Konya, which are both in the semi-arid central Anatolia region (Iyigun et al., 2013). Therefore, we hypothesized that the soil microbiota might also differ between these regions. However, the large majority of OTUs identified were present in all five provinces (Fig. 5A) and of these only a minority (86/922) were differentially abundant between regions (Fig. 5B). Furthermore, many taxa had differing levels of abundance between sampling sites from the same province (Figs 3B,5C). It has been demonstrated that soil type is a significant factor in determining microbiome composition from bulk soils (Veselovsky et al., 2025), and this could account for dissimilarities between the pair of samples from Konya and between those from Tekirdağ, which had differing soil types and characteristics (Tables 1,2). In general sample pairs from geographically close sites with the same soil type (e.g. Diy1 and Diy2, Siv1 and Siv2) were more similar to each other, but still showed some variations. This suggests that soil type and other local environmental factors play a larger role than general climate in determining the composition of the wheat rhizosphere microbiome.

One factor that clearly has a significant effect is soil pH (Fig. 3C), as several OTUs were more abundant in mildly acidic soils

(*Massilia, Rhodanobacter*) while others were enriched in neutral soils (*Arenimonas, Pseudoxanthomonas* and *Steroidobacter*). Similarly, the abundance of *Aquicella* and *Legionella* increased with macronutrient (N/P) availability, whereas low nutrient conditions favoured *Stenotrophomonas* (Fig. 6). By far the most prevalent and abundant OTU from this genus was assigned to *S. rhizophila* (Table S3), several strains of which are known to protect plants against biotic or abiotic stress (Alavi et al., 2013; Raio et al., 2023). Our data suggest that it would also be worth investigating whether this species has PGPR activity under limited nutrient conditions.

### Functional implications of changes in the rhizosphere microbiome community

One strategy for identifying putative functionally important microbial taxa is to compare the microbiome of the wheat rhizosphere with those of non-wheat crops and/or non-rhizospheric soils. Accordingly, we identified 72 OTUs that were significantly enriched in the wheat rhizosphere, although most of them were only abundant in a minority of samples (Fig. 3A,B). This sample-specific variation was clearest for the highly diverse *Pseudomonas* genus (Gomila et al., 2015), which was represented by 30 different OTUs distributed across the wheat samples. This observation supports a model in which the rhizosphere microbiome contains ecological niches that provide necessary functions, but can be filled by different, functionally similar bacterial species in different cases. Interestingly, a recent study of the effect of the rhizosphere microbiome on wheat growth reported that enrichment of *Janthinobacterium* and a Xanthomonadaceae taxon during wheat growth correlated with improved grain production (Xu et al., 2025), similar taxa were among those enriched here. Furthermore, *Pantoea agglomerans* was enriched in 5/10 of our wheat samples and was also one of the most abundant rhizosphere core microbiome taxa reported previously (Simonin et al., 2020). This species has attracted considerable interest as a possible biocontrol agent and plant growth-promoting rhizobacteria (PGPR), including in wheat (Ansari et al., 2024). However, other *P. agglomerans* isolates cause various plant diseases, and can even be opportunistic pathogens in humans (Lorenzi et al., 2022); therefore, careful strain characterization would be needed before producing it for agricultural use. The wheat-enriched taxa also include other species that are reported to act as PGPRs in different plants, such as *Variovorax paradoxus* (Chen et al., 2013), several *Pseudomonas* strains (Singh et al., 2022), and *Stenotrophomonas rhizophila* (Alavi et al., 2013).

Hub nodes, identified by their central position in co-occurrence networks (Fig. 4B,C), may include 'keystone taxa' that determine microbiome community structure; for example, in *Arabidopsis thaliana* the presence of endophytic *Albugo* sp. was demonstrated to have a significant impact on both microbiome diversity and stability (Agler et al., 2016). *Bacillus* and *Enterobacter*, which were identified as hub nodes in the positive interaction network (Fig. 4B, clusters marked with dotted lines), are known to include PGPR strains (Safdarian et al., 2020). In contrast, *Xylella* is mostly known as a re-emerging plant pathogen that can also alter microbial communities (Landa et al., 2022). Conversely *Rhodoferax* – a hub in the negative interaction network (Fig. 4C) – is reported to be enriched in genes that could make it an effective biocontrol agent (Tomita et al., 2023).

The identification of these taxa in the rhizosphere samples is circumstantial evidence that they contribute positively to crop growth and health in Turkish wheat fields. However, not all hubs represent biologically relevant 'keystones', and as noted above some enriched taxa also include potential plant pathogens; thus, isolation and characterization of individual strains is needed to determine their functional characteristics.

### Conclusion

We have confirmed that long-read amplicon sequencing is a useful strategy for meta-barcoding of environmental samples, that can be expected to become even more effective with further technological improvements. For the first time we reported on the composition of the wheat rhizosphere in diverse soils from Türkiye, allowing useful comparisons with other studies from other countries. While certain taxa such as *Pseudomonas* and *Massilia* seem to be ubiquitous in the wheat microbiome, there are many differences based on location and soil conditions. In particular, certain taxa are correlated with differences in soil pH and macronutrient availability (Table 2; Figs 3C,6). We identified a 'core Turkish wheat rhizosphere microbiome' consisting of 209 ubiquitous OTUs, while differential abundance and network analyses were used to highlight several taxa with both high prevalence and putative PGPR activity.

For future work, it is necessary to isolate these strains and characterise their interactions with each other and wheat plants individually, in order to determine their biological roles and suitability for development as biofertilizer ingredients.

## MATERIALS AND METHODS
### Sampling and soil properties

Soil samples were collected from different provinces representing Ankara, Diyarbakir, Konya, Sivas and Tekirdağ. Provincial directorates of agriculture were consulted to select sampling locations characterized by consistently high wheat yields and similar agricultural practices, summarized in Fig. 1 and Table 1. Soil collection was carried out post-harvest and as described by Simonin et al. (2020); briefly, 12 micro-regions of each field separated by at least 20 m were selected, and 1 kg of soil excavated from a depth of 0-15 cm from the surface. The 12×1 kg samples were passed through a 4 mm sieve and thoroughly mixed to give an average sample representing the entire field. Each sample was split in half, the first half being used for physical and chemical characterization at the soil analysis laboratory of SARGEM, Konya Food & Agriculture University.

In order to obtain rhizospheric soil, the second half of each sample was divided into three pots and used to cultivate durum wheat (*Triticum turgidum* ssp. *durum,* cv. 'Eminbey') in a controlled environment. Seeds obtained from the Field Crops Central Research Institute (Ankara) were surface sterilized as described previously (Kloss et al., 1984) and then germinated on sterile MS medium in the dark at 27°C for 2 days, after which six seeds were sown into each pot. Wheat plants were cultivated with a light/dark photoperiod of 16 h/8 h, ambient temperature of 22°C/18°C, 60-65% humidity for 1 month, by which time they had reached the three-leaf stage. At this stage the tallest plant from each pot was harvested, rhizospheric soil collected as described below and pooled according to sampling location (a total of ten soils), and plant heights measured from the crown to the tip of the longest leaf.

### Rhizospheric soil collection and DNA extraction

After shaking the roots to remove loose soil, the roots together with tightly bound rhizospheric soil were transferred to a 50 ml tube containing 20 ml of sterile saline solution (8.5% w/v NaCl) and agitated for 5 min on a vortex mixer. Next, the roots were removed, detached soil collected by centrifugation, and the resulting pellet frozen by submerging in liquid $N_2$ and stored at −80°C.

Total DNA was isolated from 250 mg of each soil sample (pooled from three plants) using the ZymoBIOMICS DNA Miniprep Kit (Zymo Research, Germany) according to the manufacturer's instructions. The concentration and purity of DNA isolates was verified by UV spectrophotometry and the size distribution by agarose gel electrophoresis.

For comparison with the wheat rhizospheric samples, DNA from two unrelated soils (sA7h: tomato rhizospheric soil; and KaP1: rhizospheric soil from a potato field) were isolated in the same way.

## Metabarcode library preparation and long-read sequencing

The bacterial composition of each sample was analysed by metabarcoding of the bacterial 16S rRNA gene using the two-step PCR protocol described previously (Bertolo et al., 2024) with some modifications. The first PCR used inner primers (Sentebiolab, Ankara) designed to amplify the V1-V9 regions of the rRNA gene while appending the conserved region of MinION barcode primers (underlined):

Ec27F-ONT1: 5′-TTTCTGTTGGTGCTGATATTGCAGAGTTTGAT-CCTGGCTCAGATTGA-3′

Ec1492R-ONT2: 5′-ACTTGCCTGTCGCTCTATCTTCCGATACGGY-TACCTTGTTACGACTT-3′.

PCR amplifications were carried out using the 2× LongAmp HotStart PCR kit (New England Biolabs) with the following program: initial denaturation at 94°C, 1 min; 25 amplification cycles of 94°C, 30 s, 62°C, 30 s, 65°C, 1 min 45 s; final extension, 65°C, 5 min.

The first stage amplification was optimized by amplifying serial twofold dilutions of the template DNA (to reduce the effect of PCR inhibitors co-purified with the soil DNA) until full-length PCR products (~1541 bp) were just visible by agarose gel electrophoresis. These products were then purified using AMPure XP paramagnetic beads (Beckmann Coulter, #A63881) according to the manufacturer's instructions.

For the second-stage PCR, outer primers were from the PCR Barcoding Kit (#SQK-PBK004, Oxford Nanopore Technologies), and 1 µl of the purified first-stage product was used as template in a total volume of 50 µl. As a control for contamination, 1 µl of a negative (no-template) control reaction from the first PCR was also barcoded and sequenced. PCR conditions were as above for 25 amplification cycles. The barcoded PCR products were again purified using AMPure XP beads, eluting in 10 µl of 10 mM Tris-HCl, 50 mM NaCl (pH 8.0).

The accurate concentration of PCR products from each of the 12 soil samples and the negative control were measured fluorometrically (Quant-It dsDNA assay kit, Invitrogen) and, based on the measurements, pooled in equimolar concentrations and stored at 4°C for <24 h before sequencing on a MinION Mk1B device (Oxford Nanopore Technologies) as follows. A total of 100 fmol of pooled barcoded DNA fragments were diluted to a volume of 10 µl, combined with 1 µl rapid adapters (from PCR Barcoding Kit, Oxford Nanopore Technologies), incubated for 5 min at room temperature, and then mixed with 34 µl sequencing buffer, 25.5 µl loading beads, and 4.5 µl nuclease-free water. The whole library mixture (75 µl) was loaded on to a single R9.4 flow cell (FLO-MIN106, Oxford Nanopore Technologies), and sequencing was controlled and monitored using MinKNOW software v4.2.5.

## Metagenome sequence annotation and statistical analysis

Monitoring of the sequencing run, basecalling and demultiplexing of the barcoded samples were completed by the MinKNOW software, producing adapter-trimmed read data files in FASTQ format. Output filters were selected such that low quality reads (average Q-score <8, or barcode unclassified) were saved separately and discarded. Reads passing these filters were uploaded to the Epi2ME cloud platform or Epi2ME Labs desktop agent (Oxford Nanopore Technologies), using the 'FastQ 16S Workflow' to eliminate reads <800 nt and then assign each read to its most probable taxon, using BLAST to compare reads with the NCBI 16S/18S rDNA database. Tables of OTUs quantified by read counts were also generated and used in subsequent analyses.

Microbiome data were collated and analysed in an R environment (version 4.3.2 in RStudio) using functions from the vegan (Oksanen et al., 2025) and phyloseq (McMurdie and Holmes, 2013) microbiome profiling packages. For determination of DAT, the choice of statistical methods used has been observed to have a large bearing on the number of OTUs reported as significantly different between groups. Therefore, initial investigations were made using both DESeq2 (fitting to a negative binomial distribution; Love et al., 2014) and ANCOMBC (Analysis of Composition of Microbiomes with Bias Correction; Lin and Peddada, 2020) methods. The

method that gave the most consistent results and that was used for generating the final data was the ancombc2 function, which also controls for the possibility that adding pseudocounts to the data can create false positives (option pseudo_sens=TRUE). In this study, differences between groups were only considered significant if they both passed the pseudocount test and had a q-value <0.05 (adjusted $P$-value using the Benjamini-Yekutieli method for multiple comparisons of non-independent measurements). Where analyses compared more than two experimental groups at a time, first of all the global test was carried out to identify OTUs for which at least one difference was significant under the criteria above; for these OTUs only, pairwise tests were then applied to determine which groups had differences between them, controlling for the mixed directional false discovery rate (mdFDR <0.05) using the 'Holm' family-wise error correction. For pattern analysis, taxa were evaluated using the trend function with default parameters to control for significance. Data were visualized using the VennDiagram and ComplexHeatmap packages.

Networks of co-occurring taxa were modelled and visualized in the NetCoMi R package (Peschel et al., 2021). For this purpose, only OTUs that were classified at the genus level were included. Network inference used statistical methods previously implemented in SPRING (Semi-Parametric Rank-based approach for INference in Graphical model; Yoon et al., 2019) and SpiecEasi (SParse InversE Covariance Estimation for Ecological Association Inference; Kurtz et al., 2015). Networks constructed using both methods were examined and found to have broadly similar topology. The networks shown in the Results section were derived with SpiecEasi using the following parameters: method='mb', lambda.min.ratio=0.001, nlambda=50, pulsar rep.num=20. Hub taxa were defined as those above the 90th centile within the dataset in both degree centrality (number of direct interactions) and betweenness centrality (presence on the shortest route between other points in the network).

## Acknowledgements
We would like to thank Dr Nur Koyuncu from the Field Crops Central Research Institute, Ankara, for collecting soil samples from Ankara and Prof. Dr Murat Şeker for collecting soil samples from Sivas.

## Competing interests
The authors declare no competing or financial interests.

## Author contributions
Conceptualization: Ö.A., M.G.Ş., Y.Ö.Ç., S.J.L.; Data curation: G.G., M.G.Ş., S.J.L.; Formal analysis: G.G., S.N.A., J.a.-K., S.J.L.; Funding acquisition: Ö.A., M.G.Ş., Y.Ö.Ç.; Investigation: G.G., S.N.A., J.a.-K., S.J.L.; Methodology: S.N.A., J.a.-K., V.S., S.J.L.; Project administration: M.G.Ş., Y.Ö.Ç.; Resources: M.G.Ş., V.S.; Software: S.J.L.; Supervision: Ö.A., M.G.Ş., V.S., Y.Ö.Ç., S.J.L.; Writing – original draft: G.G.; Writing – review & editing: G.G., Ö.A., M.G.Ş., V.S., Y.Ö.Ç., S.J.L.

## Funding
This project was financially supported by the Science and Technology Council of Türkiye [TÜBİTAK grant no. 121O649, to Y.Ö.Ç.] Open Access funding provided by Sabanci University. Deposited in PMC for immediate release.

## Data and resource availability
The raw sequencing data used in this project is available from the International Nucleotide Sequence Database Consortium archives (NCBI/ENA/DDBJ), BioProject accession no. PRJEB89243 (https://www.ebi.ac.uk/ena/browser/view/PRJEB89243). OTU taxonomy assignments and read counts are included in the supplementary information.

## Peer review history
The peer review history is available online at https://journals.biologists.com/bio/lookup/doi/10.1242/bio.062230.reviewer-comments.pdf

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
