## [Peer Review File · Biology Open]

Taxonomic diversity and functional adaptations indicated by the rhizospheric soil microbiome derived from Turkish wheat fields

Gülce Güralp, Sena Nur Acet, Jana al-Khodor, Özlem Akkaya, Mine Gül Şeker, Veysel Süzerer, Yelda Özden Çiftçi and Stuart James Lucas

DOI: 10.1242/bio.062230

Editor: Marie Monniaux

Review timeline

Original submission:	2 September 2025
Editorial decision:	10 September 2025
First revision received:	11 November 2025
Accepted:	25 November 2025

Original submission

First decision letter

MS ID#: bio.062230

MS Title: Taxonomic diversity and functional adaptations indicated by the rhizospheric soil microbiome derived from Turkish wheat fields

Authors: Gülce Güralp; Sena Nur Acet; Jana al-Khodor; Özlem Akkaya; Mine Gül Şeker; Veysel Süzerer; Yelda Özden Çiftçi; Stuart James Lucas

Article Type: Research Article

I have now reached a decision on the above manuscript.

The reviewer reports are shown at the bottom of this email or can be accessed, together with a copy of this decision letter, by going to:

As you will see, the reviewers raised a number of substantial criticisms that prevent me from accepting the paper at this stage.

They suggest, however, that a revised version might prove acceptable, if you can address their concerns. If you think that you can deal satisfactorily with the criticisms on revision, I would be pleased to see a revised manuscript. We would then return it to the reviewers.

More specifically, I ask you to pay a particular attention to the possible issues of reproducibility raised by both reviewers, about experimental design (number of replicates, experimental controls, justification for some practical choices) and PCR/barcoding technique.

At this stage, we also ask you to ensure your manuscript complies with our formatting guidelines. Provided you are able to fully address the referees' comments, we are positive about publication of your paper (we accept over 95% of revision submissions) and therefore hope you won't mind any extra work involved in reformatting your manuscript at this point.

Please upload both a 'clean' version of your Word file, along with a highlighted version clearly showing where you have made changes in the revised manuscript. Please avoid using 'Track changes' in Word files as these are lost in PDF conversion.

I should be grateful if you would also provide a point-by-point response detailing how you have dealt with the points raised by the reviewers in the 'Response to Reviewers' box. Please attend to all of the reviewers' comments. If you do not agree with any of their criticisms or suggestions please explain clearly why this is so.

Reviewer 1

Comments for the author

Reviewer 1: The study titled "Taxonomic diversity and functional adaptations indicated by the rhizospheric soil microbiome derived from Turkish wheat fields" provides some valuable additional information on the rhizospheric soil microbiome. However, the manuscript requires improvements to enhance its overall quality. Below, I have outlined a few points that may help strengthen the manuscript.

1. Please clarify how the adapter-trimmed FASTQ read data were generated. To my knowledge, MinKNOW does not perform adapter trimming. Additionally, from line 43, provide details on how the reads were quality-filtered (e.g., was a specific tool used for this purpose?).
2. Please provide more detailed figure legends. For example, in Figure 3A, the triangular pattern labeled as "nd" is unclear and may cause confusion.
3. Please provide a reference or explanation on how the representative taxa within the sub-cluster around *Geobacter* contribute to iron bioavailability.
4. Inferences drawn solely from interaction networks may be an overestimation. A more detailed justification is needed for the claim that *Methylotenera* and *Rhodoferrax* are limiting the growth of other deleterious bacterial strains.
5. The advantages and limitations of 16S meta-barcoding with long reads should not be placed in the Discussion section. If authors wish to mention this, two to three lines in the introduction would be sufficient.
6. This study claims that local factors play a larger role than general climate in determining the composition of the wheat rhizosphere microbiome, but studies have shown that soil microbiome shows distinct microbial community structure based on soil type, which is an intrinsic characteristic of the soil itself (PMID: 40842827). A similar pattern is observed in the present study, where soils from different regions harbor distinct microbial communities. How does this observation compare with the influence of climate factors.
7. Full-length 16S (16S-FL) amplicons does provide improved taxonomic resolution and reveal higher microbial diversity. However, this can also lead to potential overestimation, as the longer fragments may increase the number of taxa assigned without confirming whether they represent true taxa. For example, studies on marine sediment samples found no evidence that 16S-FL improved taxonomic assignment (<https://onlinelibrary.wiley.com/doi/full/10.1002/edn3.70009>). This raises concerns about whether the microbes identified in the present study truly represent the taxa present in the samples.
8. Only two samples were analyzed from each site, which limits the ability to infer the full microbial composition of the rhizospheric soil from that area. If possible, I would recommend additional sampling to better capture the broader spectrum of the microbial community.
9. The reference number (4) in line 29 is not properly placed in the main text. I recommend citing an experimental article rather than a review to support the statement made in lines 25-29.

Reviewer 2

Comments for the author

General Comments:

The study investigates the wheat rhizosphere microbiome in Turkish soils, aiming to understand the influence of soil chemistry and location, define a core microbiome, and compare results with other countries. The overall design, multi-province sampling, soil chemical analysis, and wheat cultivation under controlled conditions, is well suited to address these goals. However, the manuscript requires significant revisions to address methodological clarity, data interpretation, and presentation issues. A major revision is recommended to improve the overall quality and impact of the paper.

Specific Comments:

Reproducibility:

I. Sequencing controls (Methods): The methods do not mention negative PCR controls (no-template) or positive controls (mock microbial communities). Including these is standard in microbiome studies to check for contamination and sequencing bias. Were positive and negative controls used? If not justify the omission of the controls.

II. PCR cycle variability (Methods): The metabarcoding methods describe variable PCR cycle numbers (10-25). This could introduce technical variability and bias microbial abundance estimates. Explain how this was managed in the analysis.

III. Biological replicates (Methods): The number of biological replicates for DNA extraction needs to be specified.

IV. Independent samples vs pooling (Results): Methods indicate three pots per field (biological replicates), whereas Results appear to report one rhizosphere sample per field. Clarify how many independent rhizosphere samples were sequenced per field and whether pot-level replicates were pooled or analyzed separately

V. Figure Reporting (Results): List (n) and statistical tests in each figure/legend.

VI. Run concordance (Results/Discussion): You sequenced the libraries more than once but only analyzed the biggest/"best" run. Provide evidence of run-to-run concordance (e.g., similar clustering and high abundance correlations), or justify the run selection and discuss its impact on the conclusions.

VII. "Data not shown" (Discussion): Show evidence instead of "data not shown." The Discussion states replicate consistency but doesn't show it. Add a small plot or table with run-to-run similarity (e.g., similar sample clustering and high abundance correlations) and cite it in the Discussion.

Competence (Study design and validity):

I. Controls (Methods): Using tomato and potato rhizospheres provides useful plant-associated comparisons, but they do not replace bulk (unplanted) soil, which is important as a baseline. Were bulk (unplanted) soil controls included to establish a baseline microbiome? If not, justify the omission and discuss its impact on interpretation. Also clarify whether the tomato and potato rhizospheres were collected from the same fields/regions as the wheat samples; if not, address potential location confounding.

II. Plant sampling (Methods, line 35): Harvesting only the tallest plant from each pot may bias toward healthier plants. This approach might not capture the full variation in rhizospheric communities. Sampling all plants, or a representative subset, would provide a more balanced view. Justify your choice (rationale and any prior evidence)

III. Stage inconsistency (Abstract vs Methods): The abstract states seedlings were grown to the 4-leaf stage, while the methods section specifies the 3-leaf stage. Resolve discrepancy.

IV. Nutrient effects (Discussion): Results report clear pH/macronutrient patterns (Table 2; Fig. 3C), including taxa shifting with acidity and N/P (e.g., *Massilia*, *Rhodanobacter*, *Stenotrophomonas*). Add a short paragraph in the Discussion summarizing and interpreting these links.

V. Conclusion scope (Conclusion): Separate results from prospects by reporting descriptive findings (e.g., widespread families; 209-OTU core) as results and framing biofertilizer/PGPR potential as testable hypotheses requiring isolate characterization and plant assays, with a brief next-steps statement.

VI. Link conclusions to data (Conclusion): Add a line that clearly connects your conclusions to the pH and macronutrient patterns you reported in (Table 2; Fig. 3C)

Scholarship:

I. Abstract accuracy: To avoid overstatement and keep the abstract tightly aligned with what the study actually demonstrates, replace "essential foundation" with "provides a foundation" or "could inform the development of locally adapted bio-fertilizers" to reflect the study's correlative evidence.

II. Citations: Cite supporting and contrasting studies for the PGPR/biocontrol claims and explicitly cross-reference Fig. 4B-C when discussing hub taxa.

III. Sequence accuracy claim (Discussion): Statements that ONT V14 kits achieve ">99% per-base accuracy" require a concrete citation (vendor technical note or peer-reviewed source).

Reviewer's Responses to Questions

Experimental quality

Does each figure have the proper controls?

If 'No', please indicate reasons in Comments for Author box below.

Reviewer #1:

- Yes

Reviewer #2:

- Yes

Were the data analyzed using appropriate statistical tests?

If 'No', please indicate reasons in Comments for Author box below.

Reviewer #1:

- Yes

Reviewer #2:

- Yes

Reproducibility

Were experiments performed using adequate number of biological replicates?

If 'No', please indicate reasons in Comments for Author box below.

Reviewer #1:

- No

Reviewer #2:

- Yes

Does the methods section provide sufficient detail to permit reproducibility?

If 'No', please indicate reasons in Comments for Author box below.

Reviewer #1:

- Yes

Reviewer #2:

- No

Completeness

Are the manuscript's conclusions supported by the data?

If 'No', please indicate reasons in Comments for Author box below.

Reviewer #1:

- Yes

Reviewer #2:

- Yes

Scholarship

Do the authors cite and discuss the merits of data that would argue for and against their conclusion?

If 'No', please indicate reasons in Comments for Author box below.

Reviewer #1:

- Yes

Reviewer #2:

- Yes

Does the manuscript title & abstract accurately reflect the contents of the manuscript, without hyperbole?

If 'No', please indicate reasons in Comments for Author box below.

Reviewer #1:

- Yes

Reviewer #2:

- Yes

First revision

Author response to reviewers' comments

Dear Dr. Monniaux,

Please accept my thanks and those of the rest of the authors for your comments on our previous manuscript submission. We are also grateful to the reviewers for their constructive criticism, which

has helped us to improve the manuscript significantly in several areas. We attach the clean revised version for your consideration as requested, along with a marked up version highlighting the changes.

We have checked the formatting to conform to that of Biology Open, and responded to all of the reviewers' comments, which I have listed in detail below.

Reviewer 1:

The study titled "Taxonomic diversity and functional adaptations indicated by the rhizospheric soil microbiome derived from Turkish wheat fields" provides some valuable additional information on the rhizospheric soil microbiome. However, the manuscript requires improvements to enhance its overall quality. Below, I have outlined a few points that may help strengthen the manuscript.

Response: Many thanks for your evaluation and constructive comments, which we think have helped us to make significant improvements to the manuscript.

1. Please clarify how the adapter-trimmed FASTQ read data were generated. To my knowledge, MinKNOW does not perform adapter trimming. Additionally, from line 43, provide details on how the reads were quality-filtered (e.g., was a specific tool used for this purpose?).

Response: You are correct that MinKNOW does not trim reads when using single sequencing libraries, however with barcoded libraries adaptor trimming can be turned on as part of the demultiplexing step, as in our case. Similarly at the output step a minimum Q-score can be specified (reads below this threshold are saved in a 'fail' folder) and reads without a barcode are also saved separately (as 'unclassified'). I have pasted relevant sections of the MinKNOW user guide below.

The read length filter was applied as an initial step in the Epi2ME pipeline, we have edited the text (p8, lines 230-234) to report this more accurately.

6 Choose your barcoding options:

Barcoding options are only available when a barcoding sequencing kit or expansion has been selected. Barcoding can be switched off and performed post-sequencing.

To specify the barcoding options, click the **Edit** icon to open the barcoding options dialogue box. From here, barcode trimming and custom barcode selection can be defined.

2. Please provide more detailed figure legends. For example, in Figure 3A, the triangular pattern labeled as "nd" is unclear and may cause confusion.

Response: Thank you for highlighting the issue, we have added the following text to the legend for Fig. 3A: "For soil pH, nd = not determined (the case for the non-wheat soils)." All the legends were revised, group sizes and extra detail added wherever they were lacking (see highlighted version, p24-25).

3. Please provide a reference or explanation on how the representative taxa within the sub-cluster around *Geobacter* contribute to iron bioavailability.

Response: The following explanation has been added (p13, lines 370-372): "Geobacter and its syntrophic bacterial partners are known to have an important impact on soil chemistry, recycling organic molecules while reducing insoluble Fe(III) to soluble Fe(II) (Lovley et al, 2011). Therefore this sub-cluster may represent 'accessory' taxa that help supply wheat roots with iron in a bioavailable form."

4. Inferences drawn solely from interaction networks may be an overestimation. A more detailed justification is needed for the claim that *Methylotenera* and *Rhodofera* are limiting the growth of other deleterious bacterial strains.

Response: We agree with the reviewer's warning that it is unwise to assume too much from interaction networks alone, which was why the original text suggested that the two geni in question 'may be' limiting growth of other strains. However we have revised the relevant section to express further caution and explain our reasoning (p13, lines 376-379):

"It has been reported that negative interactions within microbiome networks play an important role in network robustness, e.g. by keeping detrimental species in check (Kajihara et al. 2025). However, inferences from correlation must be regarded as tentative, and should be confirmed by functional studies using these specific taxa."

5. The advantages and limitations of 16S meta-barcoding with long reads should not be placed in the Discussion section. If authors wish to mention this, two to three lines in the introduction would be sufficient.

Response: We have deleted most of this part of the discussion and added the following sentence to the introduction (Lines 112-115):

"Long-read meta-barcoding using Nanopore or PacBio platforms can incorporate the whole 16S gene into a single amplicon, improving resolution; however, they currently have higher per-base error rates than Illumina, while their data analysis pipelines are still under development."

We do think it is important to highlight the limitations and advantages of our technical approach for the benefit of other researchers, so have retained some comments on this at the beginning of the revised Discussion (see response to comment 7 below).

6. This study claims that local factors play a larger role than general climate in determining the composition of the wheat rhizosphere microbiome, but studies have shown that soil microbiome shows distinct microbial community structure based on soil type, which is an intrinsic characteristic of the soil itself (PMID: 40842827). A similar pattern is observed in the present study, where soils from different regions harbor distinct microbial communities. How does this observation compare with the influence of climate factors.

Response: Thank you for this excellent question, and for highlighting the informative recent study by Veselovsky et al. We have added soil type information to Table 1 and included it in the section of the discussion on 'Regional vs local environment' (p17, lines 494-501):

"It has been demonstrated that soil type is a significant factor in determining microbiome composition from bulk soils (Veselovsky et al., 2025), and this could account for dissimilarities between the pair of samples from Konya and between those from Tekirdağ, which had differing soil types and characteristics (Table 1 & 2). In general sample pairs from geographically close sites with the same soil type (e.g. Diy1 & Diy2, Siv1 & Siv2) were more similar to each other, but still showed some variations. This suggests that soil type and other local environmental factors play a larger role than general climate in determining the composition of the wheat rhizosphere microbiome."

7. Full-length 16S (16S-FL) amplicons does provide improved taxonomic resolution and reveal higher microbial diversity. However, this can also lead to potential overestimation, as the longer fragments may increase the number of taxa assigned without confirming whether they represent true taxa. For example, studies on marine sediment samples found no evidence that 16S-FL improved taxonomic assignment (<https://onlinelibrary.wiley.com/doi/full/10.1002/edn3.70009>). This raises concerns about whether the microbes identified in the present study truly represent the taxa present in the samples.

Response: Thank you for highlighting another important study for comparison and evaluation of our data. We also recognise the risk of over-identifying OTUs with the 16S-FL strategy, and have highlighted this issue and how we tried to control for it at the start of the revised discussion (p14-15):

"Alongside the advantage of generating a single amplicon from the 16S gene, the major limitation of the MinION v9 chemistry used here is its high per-base random error rate (~10%), necessitating specialized data processing (Santos et al., 2020) and introducing a risk that species diversity can be

overestimated. For example, a previous study using the same chemistry for a mock bacterial community of 15 species detected over 50 OTUs; the excess ‘species’ belonged to the genera present in the community and were only detected at low read depths, suggesting that they reflect sequencing errors and/or intra-species diversity within the 16S rRNA gene pool (Lemoine et al., 2024)..... We controlled for over-assignment of OTUs by eliminating those that had <5 reads in total or were only detected in a single sample; however, species-level assignments must still be regarded as tentative, especially for low abundance taxa.”

Reviewer 2:

General Comments:

The study investigates the wheat rhizosphere microbiome in Turkish soils, aiming to understand the influence of soil chemistry and location, define a core microbiome, and compare results with other countries. The overall design, multi-province sampling, soil chemical analysis, and wheat cultivation under controlled conditions, is well suited to address these goals. However, the manuscript requires significant revisions to address methodological clarity, data interpretation, and presentation issues. A major revision is recommended to improve the overall quality and impact of the paper.

Response: Many thanks for your positive assessment and constructive criticism. We have undertaken a full revision of the manuscript and tried to address all of the points you have raised below, which we believe has significantly improved the quality of the study.

Specific Comments:

Reproducibility:

- I. **Sequencing controls (Methods):** The methods do not mention negative PCR controls (no-template) or positive controls (mock microbial communities). Including these is standard in microbiome studies to check for contamination and sequencing bias. Were positive and negative controls used? If not justify the omission of the controls.

Response: Negative PCR controls were sequenced and gave <10 reads of the expected amplicon size, so these were eliminated during data analysis; this information has been added to the Methods. Another study using a method very similar to ours across multiple laboratories found that sequencing a mock microbial community failed to detect abnormalities arising from DNA isolation and PCR from soil samples (DOI: 10.1038/s42003-024-06594-8), an issue that is addressed at the start of the revised Discussion section (p14-15):

“Meta-barcoding is an increasingly popular method for evaluating microbiome composition, but there is not yet a consensus on which protocols offer the best balance of sensitivity and accuracy. Alongside the advantage of generating a single amplicon from the 16S gene, the major limitation of the MinION v9 chemistry used here is its high per-base random error rate (~10%), necessitating specialized data processing (Santos et al., 2020) and introducing a risk that species diversity can be overestimated. For example, a previous study using the same chemistry for a mock bacterial community of 15 species detected over 50 OTUs; the excess ‘species’ belonged to the genera present in the community and were only detected at low read depths, suggesting that they reflect sequencing errors and/or intra-species diversity within the 16S rRNA gene pool (Lemoine et al., 2024). Sequencing mock communities can identify technical biases arising during PCR amplification and library preparation; however, other studies have highlighted that they do not account for variability in DNA isolation and amplification efficiency from real soil samples (Manter et al., 2024).

In this study, we minimized inter-sample variability by maintaining consistent isolation and amplification conditions; we also sequenced PCR-negative controls to confirm that there was no bacterial contamination during library preparation.”

...and subsequent discussion of the controls used here and their limitations.

- II. **PCR cycle variability (Methods):** The metabarcoding methods describe variable PCR cycle numbers (10-25). This could introduce technical variability and bias microbial abundance estimates. Explain how this was managed in the analysis.

Response: Thank you for highlighting this oversight; the PCR cycle numbers were varied during protocol optimization but the final samples for sequencing were all amplified for 25 cycles. The text has been corrected (p7, line 203). As described in the same section, the initial DNA concentrations from each soil sample were adjusted to give equivalent product concentrations at the end of the first PCR, to minimize amplification bias that might arise from variable PCR efficiencies.

- III. **Biological replicates (Methods):** The number of biological replicates for DNA extraction needs to be specified.

Response: 3 biological replicates were pooled at the point of DNA extraction, this is now explicitly stated in the text (Lines 176-177, 186).

- IV. **Independent samples vs pooling (Results):** Methods indicate three pots per field (biological replicates), whereas Results appear to report one rhizosphere sample per field. Clarify how many independent rhizosphere samples were sequenced per field and whether pot-level replicates were pooled or analyzed separately.

Response: 3 biological replicates were pooled at the point of DNA extraction, so there is one sequenced sample per field; this is now explicitly stated in the text (Lines 176-177, 186). As we were developing the methods and were not certain what the level of sequencing output would be, we pooled the biological replicates to ensure that the maximum diversity was represented in the final dataset; although obviously this reduced the statistical power of the study.

- V. **Figure Reporting (Results):** List (n) and statistical tests in each figure/legend.

Response: This information has been added to all the figure legends, along with extra detail as requested by the other reviewer (see highlighted version of the revised text, p24-25).

- VI. **Run concordance (Results/Discussion):** You sequenced the libraries more than once but only analyzed the biggest/“best” run. Provide evidence of run-to-run concordance (e.g., similar clustering and high abundance correlations), or justify the run selection and discuss its impact on the conclusions.

Response: We have now added Supplementary figures displaying the clustering and abundance correlations for all 3 runs, demonstrating consistency in results; they are now explicitly referred to in the relevant Results section (p11, lines 296-302):

“All runs gave similar results in terms of the abundance of OTUs detected and clustering of samples (see Figures S1 & S2), even though there was some variation between runs and between samples in sequencing depth. In all cases the peak read length was ~1.5kb, demonstrating that the 16S (V1-V9) amplicon was sequenced mostly as a single read. Run 3 was the most productive run (1.23 million passed reads, more than twice as many as the other runs) and also had the most even distribution of read counts between the 12 samples, and so was used for the analyses below.”

- VII. **“Data not shown” (Discussion):** Show evidence instead of “data not shown.” The Discussion states replicate consistency but doesn’t show it. Add a small plot or table with run-to-run similarity (e.g., similar sample clustering and high abundance correlations) and cite it in the Discussion.

Response: This sentence was removed from the revised discussion, but the question of replicate consistency is addressed in the revised Results (see response to the previous comment).

Competence (Study design and validity):

- I. **Controls (Methods):** Using tomato and potato rhizospheres provides useful plant-associated comparisons, but they do not replace bulk (unplanted) soil, which is important as a baseline. Were bulk (unplanted) soil controls included to establish a baseline microbiome? If not, justify the omission and discuss its impact on interpretation. Also clarify whether the tomato and potato rhizospheres were collected from the same fields/regions as the wheat samples; if not, address potential location confounding.

Response: We agree that bulk (unplanted) soils would be a useful baseline, but unfortunately were not included in our original study design due to resource limitations - as each field would be different, it would have doubled the number of samples to be sequenced. Also, our primary goal was to identify candidate taxa involved specifically in promoting wheat growth, so we prioritized comparison with rhizospheric soil from other crops. The tomato and potato rhizospheres were not collected from the same location, as they are not typically cultivated close to each other in Türkiye. These potential confounding issues are addressed in the revised discussion, (p15, lines 445-451):

“Based on previous studies, [rhizospheric soil] is expected to contain a subset of the microbial diversity in the surrounding bulk soil, enriched for certain taxa such as Proteobacteria (Ling et al., 2022), which was also the most abundant phylum detected in our dataset. We highlighted differences between the wheat rhizospheric microbiome and 2 rhizosphere samples collected from other crops grown in different locations; while some differences may be due to location-specific variation, 72 OTUs were consistently more abundant in the wheat rhizosphere samples than controls (Figure 3B).

- II. **Plant sampling (Methods, line 35):** Harvesting only the tallest plant from each pot may bias toward healthier plants. This approach might not capture the full variation in rhizospheric communities. Sampling all plants, or a representative subset, would provide a more balanced view. Justify your choice (rationale and any prior evidence)

Response: This part is revised as follows (p6, line 176-177) “At this stage the tallest plant from each pot was harvested, rhizospheric soil collected as described below and pooled according to sampling location (a total of 10 soils)...” We took plant height as an indicator of better growth of the seedling, and preferred to select these due to our aim of identifying microbes with plant growth promoting activities. However, the tallest plant was close in both proximity and height to other plants in the pot and shared the same soil as there were at least 5 plants per pot; therefore their microbial communities are expected to be very similar. As the soil in each pot was already obtained by pooling the soil samples from each location, it is expected that between-pot variation in rhizospheric communities would be relatively limited.

- III. **Stage inconsistency (Abstract vs Methods):** The abstract states seedlings were grown to the 4-leaf stage, while the methods section specifies the 3-leaf stage. Resolve discrepancy.

Response: Thank you for picking up this error, the abstract was corrected as seedlings were grown to the 3-leaf stage.

- IV. **Nutrient effects (Discussion):** Results report clear pH/macronutrient patterns (Table 2; Fig. 3C), including taxa shifting with acidity and N/P (e.g., *Massilia*, *Rhodanobacter*, *Stenotrophomonas*). Add a short paragraph in the Discussion summarizing and interpreting these links.

Response: We have collated and expanded the comments on these results in the Discussion on page 17, lines 503-511:

“One factor that clearly has a significant effect is soil pH (Fig. 3C), as several OTUs were more abundant in mildly acidic soils (*Massilia*, *Rhodanobacter*) while others were enriched in neutral soils (*Arenimonas*, *Pseudoxanthomonas* & *Steroidobacter*). Similarly, the abundance of *Aquicella* & *Legionella* increased with macronutrient (N/P) availability, whereas low nutrient conditions

favoured *Stenotrophomonas* (Fig. 6). By far the most prevalent and abundant OTU from this genus was assigned to *S. rhizophila* (Table S3), several strains of which are known to protect plants against biotic or abiotic stress (Alavi et al., 2013; Raio et al., 2023). Our data suggest that it would also be worth investigating whether this species has PGPR activity under limited nutrient conditions.”

- V. **Conclusion scope (Conclusion):** Separate results from prospects by reporting descriptive findings (e.g., widespread families; 209-OTU core) as results and framing biofertilizer/PGPR potential as testable hypotheses requiring isolate characterization and plant assays, with a brief next-steps statement.

Response: We completely agree with the reviewer (isolation and characterization of these strains is the next phase of our ongoing project). The end of the conclusion (p19) has been revised as suggested:

“We identified a ‘core Turkish wheat rhizosphere microbiome’ consisting of 209 ubiquitous OTUs, while differential abundance and network analyses were used to highlight several taxa with both high prevalence and putative PGPR activity.

For future work, it is necessary to isolate these strains and characterise their interactions with each other and wheat plants individually, in order to determine their biological roles and suitability for development as biofertilizer ingredients.”

- VI. **Link conclusions to data (Conclusion):** Add a line that clearly connects your conclusions to the pH and macronutrient patterns you reported in (Table 2; Fig. 3C)

Response: We have added the following sentence (lines 562-3):

“...there are many differences based on location and soil conditions. In particular, certain taxa are correlated with differences in soil pH and macronutrient availability (Table 2; Figs. 3C & 6).”

Scholarship:

- I. **Abstract accuracy:** To avoid overstatement and keep the abstract tightly aligned with what the study actually demonstrates, replace “essential foundation” with “provides a foundation” or “could inform the development of locally adapted bio-fertilizers” to reflect the study’s correlative evidence.

Response: We have softened the claim as per the reviewer’s first suggestion.

- II. **Citations:** Cite supporting and contrasting studies for the PGPR/biocontrol claims and explicitly cross-reference Fig. 4B-C when discussing hub taxa.

Response: We have revised and expanded the relevant section of the discussion as follows (p18):

“However, other *P. agglomerans* isolates cause various plant diseases, and can even be opportunistic pathogens in humans (Lorenzi et al., 2022); therefore careful strain characterization would be needed before producing it for agricultural use.”

.....

“Hub nodes, identified by their central position in co-occurrence networks (Fig 4B & C), may include ‘keystone taxa’ that determine microbiome community structure; for example, in *Arabidopsis thaliana* the presence of endophytic *Albugo* sp. was demonstrated to have a significant impact on both microbiome diversity and stability (Agler et al., 2016). *Bacillus* and *Enterobacter*, which were identified as hub nodes in the positive interaction network (Fig. 4B - clusters marked with dotted lines), are known to include PGPR strains (Safdarian et al., 2020). In contrast, *Xylella* is mostly known as a re-emerging plant pathogen that can also alter microbial communities (Landa et al., 2022).”

“However, not all hubs represent biologically relevant ‘keystones,’ and as noted above some enriched taxa also include potential plant pathogens; thus isolation and characterization of individual strains is needed to determine their functional characteristics.”

- III. **Sequence accuracy claim (Discussion):** Statements that ONT V14 kits achieve “>99% per-base accuracy” require a concrete citation (vendor technical note or peer-reviewed source).

Response: During revision we decided that the comment about ONT v14 kits was not directly relevant to our current study, so have deleted it (although here is the DOI of an example validating the 99% accuracy claim for gut microbiome samples: 10.1038/s41467-024-51929-y)

Second decision letter

MS ID#: bio.062230R1

MS Title: Taxonomic diversity and functional adaptations indicated by the rhizospheric soil microbiome derived from Turkish wheat fields

Authors: Gülce Güralp; Sena Nur Acet; Jana al-Khodor; Özlem Akkaya; Mine Gül Şeker; Veysel Süzerer; Yelda Özden Çiftçi; Stuart James Lucas

Article Type: Research Article

I am happy to tell you that your manuscript has been accepted for publication in Biology Open, pending our standard publication integrity checks. It was accepted on 25th November 2025.

However, it was pointed out by Reviewer 1 that you did not answer to his/her comments number 8 and 9. Point number 8 was a suggestion to sample additional sites for the study, but considering the amount of work this would represent, and the fact that the conclusions are already supported by the data, I don't request that this point is specifically addressed. In point number 9, it was highlighted that reference number 4 was misplaced, and should be replaced by a research article rather than a review. Please take this into account for the final version of the manuscript.